# Aggressive Q-Learning with Ensembles: Achieving Both High Sample Efficiency and High Asymptotic Performance

## Abstract

Recently, Truncated Quantile Critics (TQC), using distributional representation of critics, was shown to provide state-of-the-art asymptotic training performance on all environments from the MuJoCo continuous control benchmark suite. Also recently, Randomized Ensemble Double Q-Learning (REDQ), using a high update-to-data ratio and target randomization, was shown to achieve high sample efficiency that is competitive with state-of-the-art model-based methods. In this paper, we propose a novel model-free algorithm, Aggressive Q-Learning with Ensembles (AQE), which improves the sample-efficiency performance of REDQ and the asymptotic performance of TQC, thereby providing overall state-of-the-art performance during all stages of training. Moreover, AQE is very simple, requiring neither distributional representation of critics nor target randomization.

## 1 Introduction

Off-policy Deep Reinforcement Learning algorithms aim to improve sample efficiency by reusing past experience. A number of off-policy Deep RL algorithms have been proposed for control tasks with continuous state and action spaces, including Deep Deterministic Policy Gradient (DDPG), Twin Delayed DDPG (TD3) and Soft Actor Critic (SAC) (Lillicrap et al., 2016; Fujimoto et al., 2018; Haarnoja et al., 2018a;b). TD3 introduced clipped double-Q learning, and was shown to be significantly more sample efficient than popular on-policy methods for a wide range of MuJoCo benchmarks. Soft Actor Critic (SAC) has similar off-policy structures with clipped double-Q learning, but it also employs maximum entropy reinforcement learning. SAC was shown to provide excellent sample efficiency and asymptotic performance in a wide-range of MuJoCo environments, including the high-dimensional Humanoid environment for which both DDPG and TD3 perform poorly.

More recently, Kuznetsov et al. (2020) proposed Truncated Quantile Critics (TQC), a model-free algorithm which includes distributional representations of critics, truncation of critics prediction, and ensembling of multiple critics. Instead of the usual modeling of the Q-function of the expectation of return, TQC approximates the distribution of the return random variable conditioned on the state and action. By dropping several of the top-most "atoms" and varying the number of dropped atoms of the return distribution approximation, TQC can control the over-estimation bias. TQC's asymptotic performance (that is after a long period of training) was shown to be better than that of SAC on the continuous control MuJoCo benchmark suite, including a 25% improvement on the most challenging Humanoid environment. However, TQC is not sample efficient in that it generally requires a large number of samples to reach even moderate performance levels.

Chen et al. (2021) proposed Randomized Ensembled Double Q-learning(REDQ), a model-free algorithm which includes a high Update-To-Data (UTD) ratio, an ensemble of Q functions, and in-target minimization across a random subset of Q functions from the ensemble. Using a UTD ratio much larger than one, meaning that several gradient steps are taken for each environment interaction, improves sample efficiency, while the ensemble and in-target minimization allows the algorithm to maintain stable and near-uniform bias under the high UTD ratio. The algorithm was shown to attain much better performance than SAC at the early stage of training, and to match or improve the sample-efficiency of the state-of-the-art model-based algorithms for the MuJoCo benchmarks. How-

ever, although REDQ is highly sample efficient for early-stage training, its asymptotic performance is significantly below that of TQC.

Is it possible to design a simple, streamlined model-free algorithm which can achieve REDQ's high sample efficiency in early-stage training and also achieve TQC's high asymptotic performance in late state training? In this paper, we achieve this goal with a new model-free algorithm, Aggressive Q-Learning with Ensembles (AQE). Like TQC and REDQ, AQE uses an ensemble of Q-functions, and like REDQ it uses a UTD ratio $> 1$. However AQE is very simple, requiring neither distributional representation of critics as in TQC nor target randomization and double-Q learning as in REDQ. AQE controls over estimation bias and the standard deviation of the bias by varying the number of ensemble members $N$ and the number of ensembles $K \leq N$ that are kept when calculating the targets.

Through carefully designed experiments, we provide a detailed analysis of AQE. We first show that for the MuJoCo benchmark, AQE provides state-of-the-art performance, surpassing the performance of SAC, REDQ, and TQC at *all stages of training*. We perform an ablation study, and show that AQE is robust to choices of hyperparameters: AQE can work well with small ensembles consisting of 10-20 ensemble members, and performance does not vary significantly with small changes in the keep parameter $K$. We show that that AQE performs better than several variations, including using the median of all ensemble members and removing the most extreme minimum and maximum outlier in the targets. In order to improve computational time, we also consider different multi-head architectures for the ensemble of critics: consistent with the supervised convolutional network literature, we find that a two-head architecture not only reduces computational time but can actually improve performance for some environments. Additionally, we show that AQE continues to outperform SAC and TQC even when these algorithms are made aggressive with a UTD $\gg 1$.

To ensure our comparisons are fair, and to ensure our results are reproducible (Henderson et al., 2018; Islam et al., 2017; Duan et al., 2016), we provide open source code[1]. For all algorithmic comparisons, we use the the authors' code.

## 2 ADDITIONAL RELATED WORK

Overestimation bias due to in target maximization in Q-learning can significantly slow learning (Thrun & Schwartz, 1993). For tabular Q-learning, van Hasselt (2010) introduced Double Q-Learning, and showed that it removes the overestimation basis and in general leads to an under-estimation bias. Hasselt et al. (2016) showed that adding Double Q-learning to deep-Q networks can have a similar effect, leading to a major performance boost for the Atari games benchmark. As stated in the Introduction, for continuous-action spaces, TD3 and SAC address the overestimation bias using clipped-double Q-learning, which brings significant performance improvements (Fujimoto et al., 2018; Haarnoja et al., 2018a;b).

As mentioned in the Introduction, Kuznetsov et al. (2020) control the over-estimation bias by estimating the distribution of the return random variable, and then by dropping several of the top-most "atoms" from the estimated distribution. The distribution estimate is based on a methodology developed in Bellemare et al. (2017); Dabney et al. (2018a;b), which employs an asymmetric Huber loss function to minimize the Wasserstein distance between the neural network output distribution and the target distribution. In this paper, in order to counter over-estimation bias, we also drop the top-most estimates, although we do so solely with an ensemble of Q-function mean estimators rather than with an ensemble of the more complex distributional models employed in (Kuznetsov et al., 2020).

It is well-known that using ensembles can improve the performance of DRL algorithms (Faußer & Schwenker, 2015; Osband et al., 2016; Lee et al., 2021). For Q-learning based methods, Anschel et al. (2017) use the average of multiple Q estimates to reduce variance. Lan et al. (2020) introduced Maxmin Q-learning, which uses the minimum of all the Q-estimates rather than the average. Agarwal et al. (2020) use Random Ensemble Mixture (REM), which employs a random convex combination of multiple Q estimates.

---

[1]https://anonymous.4open.science/r/Aggressive-Q-Learning-with-Ensembles-0FC0/

Model-based methods often attain high sample efficiency by using a high UTD ratio. In particular, Model-Based Policy Optimization (MBPO) (Janner et al., 2019) uses a large UTD ratio of 20-40. Compared to Soft-Actor-Critic (SAC), which is model-free and uses a UTD of 1, MBPO achieves much higher sample efficiency in the OpenAI MuJoCo benchmark (Todorov et al., 2012; Brockman et al., 2016). REDQ (Chen et al., 2021), a model-free algorithm, also successfully employs a high UTD ratio to achieve high sample efficiency.

## 3 ALGORITHM

We propose Aggressive Q-learning with Ensembles (AQE), a simple model-free algorithm which provides state-of-the-art performance for the MuJoCo benchmark for both early and late stage of training. The pseudo-code is shown in Algorithm 1. As is the case with most off-policy continuous-control algorithms, AQE has a single actor (policy network) and multiple critics (Q-function networks), and employs Polyak averaging of the target parameters to enhance stability. Building on this algorithmic base, it also employs an update-to-data ratio $G > 1$, an ensemble of $N \geq 3$ Q-functions (rather than just two as in TD3 and SAC), and targets that average all the Q-functions excluding the Q-functions with the highest $N - K$ values. For exploration, it uses entropy maximization as in SAC, although it could easily incorporate alternative exploration schemes.

The design of AQE borrows components from TQC and REDQ, but also excludes components from those algorithms. Like TQC, AQE does not employ clipped double Q-learning to control over estimation bias, but instead drops the highest Q values when calculating the targets. Like REDQ, AQE employs a UTD $> 1$ to improve sample efficiency during training. However, unlike TQC, AQE uses a UTD $> 1$, and does not employ distributional representations of critics but instead the simpler Q-function expectation estimates. Unlike REDQ which employs two randomly chosen ensemble members when calculating the target, AQE does not use target randomization and uses most of the ensemble members when calculating the target. As discussed in the theory Section 5, this provides AQE more flexibility through the choice of the $N$. The resulting algorithm is not only simple and streamlined, but also provides state-of-the art performance.

AQE has three key hyperparameters, $G$, $N$, and $K$. If we set $N = 2$, $K = 1$ and $G = 1$, AQE is simply the underlying off-policy algorithm such as SAC. When $N > 2$, $K = 1$ and $G = 1$, then AQE becomes similar to, but not equivalent to, Maxmin Q-learning (Lan et al., 2020).

### 3.1 MULTI-HEAD ENSEMBLE ARCHITECTURE

AQE uses an ensemble of Q networks (as does REDQ and TQC). Employing multiple networks, one for each Q-function output, can be expensive in terms of computation and memory. In order to reduce the computation and memory requirements, we also consider multi-head architectures for generating multiple Q-function outputs. Instead of each network providing a single Q-estimate output, we consider $N$ separate Q networks each with $h$ heads, providing a total of $h \cdot N$ estimates. The $h$ heads from one network share all of the layers except the final fully-connected layer. In practice, we have found $h = 2$ heads works well for AQE, consistent with work in ensembles of convolutional neural networks for computer vision tasks (Lee et al., 2015). When properly sharing low level weights, multi-headed networks may not only retain the performance of full ensembles, but can sometimes outperform them. We conduct ablation studies on the multi-head architecture in AQE in Section 4.2.

## 4 EXPERIMENTS RESULTS

We provide experimental results in this section for AQE, TQC, REDQ and SAC for the five most challenging MuJoCo environments, namely, Hopper, Walker2d, HalfCheetah, Ant and Humanoid. To make a fair comparison, the TQC, REDQ and SAC results are reproduced using the authors' open source code, and use the same network sizes and hyperparameters reported in their papers. In particular, TQC employs 5 critic networks with 25 distributional samples for a total of 125 atoms. TQC drops 5 atoms per critic for Hopper, 0 atoms per critic for Half Cheetah, and 2 atoms per critic for Walker, Ant, and Humanoid. For REDQ, we also use the authors' suggested values of $N = 10$ and $M = 2$, where $M$ is the number ensemble members used in the target calculation.

---

**Algorithm 1** Aggressive Q-Learning with Ensembles

---

1: Initial policy parameters $\theta$, $N$ Q-function parameters $\phi_i, i = 1,\ldots, N$, empty replay buffer $\mathcal{D}$.
   Set target parameters $\phi_{\text{targ},i} \leftarrow \phi_i$ for $i = 1, 2,\ldots, N$.
2: **repeat**
3:     Take one action $a_t \sim \pi_\theta(\cdot|s_t)$. Observe reward $r_t$, new state $s_{t+1}$.
4:     Add data to replay buffer: $\mathcal{D} \leftarrow \mathcal{D} \cup \{(s_t, a_t, r_t, s_{t+1})\}$
5:     **for** $G$ updates **do**
6:         Randomly sample a mini-batch $B = \{(s, a, r, s')\}$ from $\mathcal{D}$.
7:         **for** each $(s, a, r, s') \in B$ **do**
8:             Sample $\tilde{a}' \sim \pi_\theta(\cdot|s')$.
9:             Determine the $K$ indices from $i = 1, \ldots, N$ that minimize $Q_{\text{target},i}(s', \tilde{a}')$.
10:            Compute the Q target $y$:

$$y(s, a) = r + \gamma \left( \frac{1}{K} \sum_{i \in K} Q_{\phi_{\text{targ},i}}(s', \tilde{a}') - \alpha \log \pi_\theta(\tilde{a}'|s') \right)$$

11:         **for** $i = 1, \ldots, N$ **do**
12:            Update $\phi_i$ with gradient descent using

$$\nabla_{\phi_i} \frac{1}{|B|} \sum_{(s,a,r,s') \in B} (Q_{\phi_i}(s, a) - y(s, a))^2$$

13:            Update target networks with $\phi_{\text{targ},i} \leftarrow \rho\phi_{\text{targ},i} + (1-\rho)\phi_i$
14:     Update policy parameters $\theta$ with gradient ascent using

$$\nabla_\theta \frac{1}{|B|} \sum_{s \in B} \left( \frac{1}{N} \sum_{i=1}^{N} Q_{\phi_i}(s, \tilde{a}_\theta(s)) - \alpha \log \pi_\theta(\tilde{a}_\theta(s)|s) \right) \qquad \tilde{a}_\theta(s) \sim \pi_\theta(\cdot|s)$$

15: **until** Convergence

---

The REDQ paper uses $G = 20$ for the update-to-data ratio, and provides results for up to 300K environment interactions. Using such a high value for $G$ is computationally infeasible in our our experimental setting, since we use 3 million environment interactions for Ant and Humanoid in order to investigate asymptotic performance as well early-stage sample efficiency. In the experiments reported here, we use a value of $G = 5$ for both REDQ and AQE.

For AQE, we use 10 Q-networks each with 2 heads, producing 20 Q-values for each input. The AQE networks are the same size as those in the REDQ paper. AQE keeps 10 out of 20 values for Hopper, all 20 values for half-Cheetah, and 16 out of 20 values for Walker, Ant and Humanoid.

Figure 1 shows the training curves for AQE, TQC, REDQ, and SAC. For each algorithm, we plot the average return of 5 independent trials as the solid curve, and plot the standard deviation across 5 seeds as the shaded region. For each environment, we train each algorithm for exactly the same number of environment interactions as done in the SAC paper. We use the same evaluation protocol as in the TQC paper. Specifically, after every epoch, we run ten test episodes with the current policy, record the undiscounted sum of all the rewards in each episode and take the average of the sums as the performance. A more detailed discussion on hyperparameters and implementation details is given in the Appendix.

We see from Figure 1 that AQE is the only algorithm that beats SAC in all five environments during all stages of training. Moreover, it typically beats SAC by a very wide margin. Table 1 shows that, when averaged across the five environments, AQE achieves SAC asymptotic performance approximately 3x faster than SAC and 2x faster than REDQ and TQC. As seen from Figure 1 and Table 2, in the early stages of training, AQE matches the excellent performance of REDQ in all five environments, and both algorithms are much more sample efficient than SAC and TQC. As seen from Figure 1 and Table 3, in late-stage training, AQE always matches or beats all other algorithms, except for Humanoid, where TQC is about 10% better. Table 3 shows that, when averaged across all five environments, AQE's asymptotic performance is 26%, 22%, and 6% higher than SAC, REDQ, and TQC, respectively.

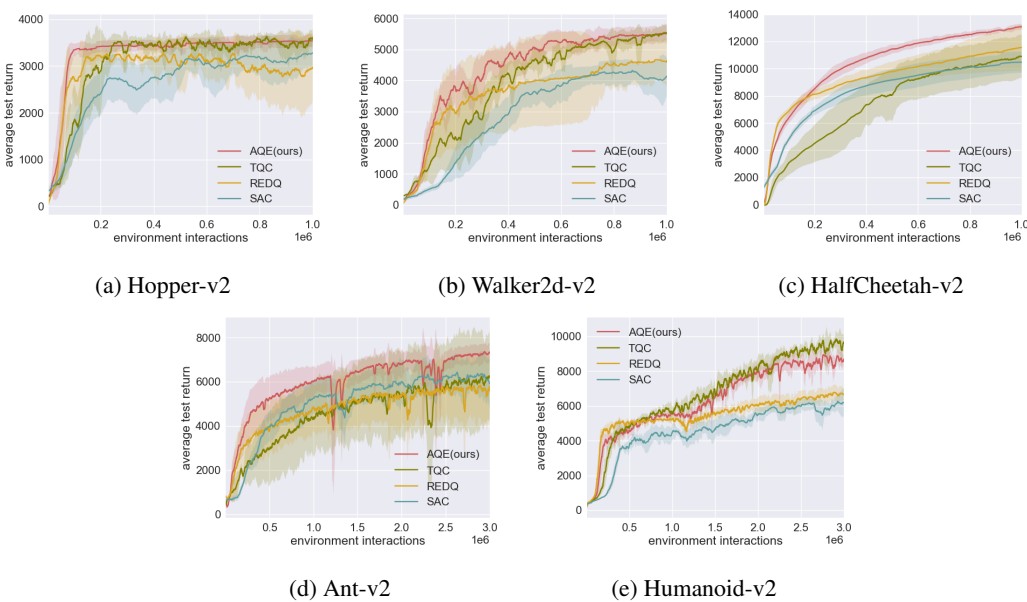

(a) Hopper-v2        (b) Walker2d-v2        (c) HalfCheetah-v2

(d) Ant-v2        (e) Humanoid-v2

Figure 1: AQE versus TQC, REDQ and SAC. AQE is the only algorithm that beats SAC in all five environments during all stages of training, and it typically beats SAC by a wide margin.

| Performance | SAC | TQC | REDQ | AQE | AQE/SAC | AQE/TQC | AQE/REDQ |
|---|---|---|---|---|---|---|---|
| Hopper at 3000 | 506K | 184K | 136K | 77K | 6.57 | 2.39 | 1.77 |
| Walker2d at 4000 | 631K | 371K | 501K | 277K | 2.28 | 1.34 | 1.81 |
| HalfCheetah at 10000 | 763K | 737K | 552K | 304K | 2.51 | 2.42 | 1.82 |
| Ant at 5500 | 1445K | 1759K | 1749K | 632K | 2.29 | 2.78 | 2.77 |
| Humanoid at 6000 | 2469K | 1043K | 1862K | 1345K | 1.84 | 0.78 | 1.38 |
| Average | - | - | - | - | 3.10 | 1.94 | 1.91 |

Table 1: Sample efficiency comparison of SAC, TQC, REDQ and AQE. The numbers show the amount of data collected when the specified performance level is reached (roughly corresponding to 90% of SAC's final performance). The last three columns show how many times AQE is more sample efficient than SAC, TQC and REDQ in reaching that performance level.

## 4.1 FIXED HYPERPARAMETER ACROSS ENVIRONMENTS

Following the TQC paper, in Figure 1 we used different drop atoms for TQC for the different environments. To make the comparison fair, we also used different keep values $K$ for AQE for the different environments. In Figure 2, we repeat the experiment on the five MuJoCo environments, but now use the same hyperparameter values across environments for TQC (drop two atoms per network) and AQE ($K = 16$). These choices of fixed hyper-parameters appear to give the best overall performance for the two algorithms. We report detailed early-stage and late-stage performance comparisons of all algorithms in Table 4 and Table 5.

We can see from the results that with fixed hyperparamters, the conclusions for AQE remain largely unchanged, except for Hopper, where REDQ becomes the strongest algorithm. Table 4 shows that when averaging performance across environments, AQE still matches the high sample efficiency of REDQ during the early stages of training. Furthermore, Table 5 shows that, on average, AQE's asymptotic performance is still 16%, 11% and 9% higher than SAC, REDQ and TQC, respectively.

## 4.2 ABLATIONS

In this section, we use ablations to provide further insight into AQE. We focus on the Ant environment, and run the experiments up to 1M time steps. (In the Appendix we provide ablations for the other four environments.) As in the REDQ paper, we consider not only performance but normalized

| Amount of data | SAC | TQC | REDQ | AQE | AQE/SAC | AQE/TQC | AQE/REDQ |
|---|---|---|---|---|---|---|---|
| Hopper at 100K | 1456 | 1807 | 2747 | 3345 | 2.30 | 1.85 | 1.22 |
| Walker2d at 100K | 501 | 1215 | 1810 | 2150 | 4.29 | 1.77 | 1.19 |
| HalfCheetah at 100K | 3055 | 4801 | 6876 | 6378 | 2.09 | 1.33 | 0.93 |
| Ant at 250K | 2107 | 2344 | 3279 | 4153 | 1.97 | 1.77 | 1.27 |
| Humanoid at 250K | 1094 | 3038 | 4535 | 3973 | 3.63 | 1.31 | 0.88 |
| Average at early stage | - | - | - | - | 2.86 | 1.61 | 1.10 |

Table 2: Early-stage performance comparison of SAC, TQC, REDQ and AQE. The numbers show the performance achieved when the specific amount of data is collected. On average, AQE performs 2.9 times better than SAC, 1.6 times better than TQC and 1.1 times better than REDQ.

| Amount of data | SAC | TQC | REDQ | AQE | AQE/SAC | AQE/TQC | AQE/REDQ |
|---|---|---|---|---|---|---|---|
| Hopper at 1M | 3282 | 3612 | 2954 | 3541 | 1.08 | 0.98 | 1.20 |
| Walker2d at 1M | 4134 | 5532 | 4637 | 5517 | 1.33 | 1.00 | 1.19 |
| HalfCheetah at 1M | 10475 | 10887 | 11562 | 13093 | 1.25 | 1.20 | 1.13 |
| Ant at 3M | 5903 | 6186 | 5785 | 7345 | 1.24 | 1.19 | 1.27 |
| Humanoid at 3M | 6177 | 9593 | 6649 | 8680 | 1.41 | 0.91 | 1.31 |
| Average at late stage | - | - | - | - | 1.26 | 1.06 | 1.22 |

Table 3: Late-stage performance comparison of SAC, TQC, REDQ and AQE. The numbers show the performance achieved when the specific amount of data is collected. The last three columns show the ratio of AQE performance compared to SAC, TQC and REDQ performance. On average, during late-stage training, AQE performs 1.26 times better than SAC, 1.06 times better than TQC and 1.22 times better than REDQ.

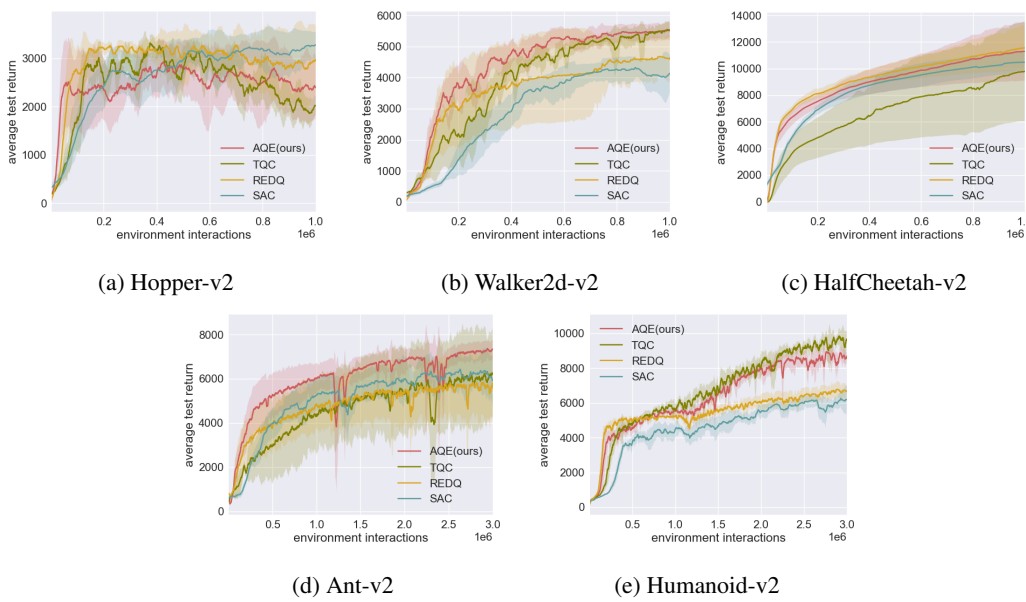

Figure 2: Performance for AQE and TQC using same hyper-parameters across the five environments. AQE uses $K = 16$ and TQC uses atoms = 2 per critic.

bias and standard deviation of normalized bias as defined by the REDQ authors. We first look at how the ensemble size $N$ affects AQE. The first row in Figure 3 shows AQE with $N$ equal to 2, 5, 10 and 15, with two heads for each Q network, and the percentage of kept Q-values unchanged. As the ensemble size $N$ increases, we generally obtain a more stable average bias, a lower std of bias, and stronger performance. When trained with high UTD value, a relatively small ensemble size, for example, $N = 5$, can greatly reduce bias accumulation, resulting in much stronger performance. This experimental finding is consistent with the results in Theorem 1 in Section 5.

| Amount of data | SAC | TQC | REDQ | AQE | AQE/SAC | AQE/TQC | AQE/REDQ |
|---|---|---|---|---|---|---|---|
| Hopper at 100K | 1456 | 1719 | 2747 | 2294 | 1.58 | 1.33 | 0.84 |
| Walker2d at 100K | 501 | 1215 | 1810 | 2150 | 4.29 | 1.77 | 1.19 |
| HalfCheetah at 100K | 3055 | 3594 | 6876 | 6325 | 2.10 | 1.76 | 0.92 |
| Ant at 250K | 2107 | 2344 | 3279 | 4153 | 1.97 | 1.77 | 1.27 |
| Humanoid at 250K | 1094 | 3038 | 4535 | 3973 | 3.63 | 1.31 | 0.88 |
| Average at early stage | - | - | - | - | 2.71 | 1.59 | 1.02 |

Table 4: Early-stage performance comparison of SAC, TQC, REDQ and AQE when AQE and TQC when using the same hyperparameters across the environments. On average, AQE performs 2.71 times better than SAC, 1.59 times better than TQC and 1.02 times better than REDQ.

| Amount of data | SAC | TQC | REDQ | AQE | AQE/SAC | AQE/TQC | AQE/REDQ |
|---|---|---|---|---|---|---|---|
| Hopper at 1M | 3282 | 2024 | 2954 | 2404 | 0.73 | 1.19 | 0.81 |
| Walker2d at 1M | 4134 | 5532 | 4637 | 5517 | 1.33 | 1.00 | 1.19 |
| HalfCheetah at 1M | 10475 | 9792 | 11562 | 11293 | 1.08 | 1.15 | 0.98 |
| Ant at 3M | 5903 | 6186 | 5785 | 7345 | 1.24 | 1.19 | 1.27 |
| Humanoid at 3M | 6177 | 9593 | 6649 | 8680 | 1.41 | 0.91 | 1.31 |
| Average at late stage | - | - | - | - | 1.16 | 1.09 | 1.11 |

Table 5: Late-stage performance comparison of SAC, TQC, REDQ and AQE when AQE and TQC when using the same hyperparameters across the environments. On average, AQE performs 16% better than SAC, 9% better than TQC and 11% times better than REDQ.

The second row in Figure 3 shows the how the keep parameter can affect the algorithm's performance: under the same high UTD value, as $K$ decreases, the average normalized Q bias goes from over-estimation to under-estimation. Consistent with the theoretical result in Theorem 1, by decreasing $K$ we lower the average bias. When $K$ becomes too small, the Q estimate becomes too conservative and starts to have negative bias, which makes learning difficult. We see that $K = 16$ has an average bias closest to 0 and also a consistently small std of bias. These results are similar for the other four environments, as shown in the Appendix.

The third row in Figure 3 shows results for variants of the target computation methods. The Median curve uses the median value of all the Q estimates in the ensemble to compute the Q target. The RemoveMinMax curve drops the minimum and maximum values of all the Q estimates in the ensemble to compute the Q target. We see that these two variants give larger positive Q bias values.

We also considered different combinations of ensemble size $N$ and the number of multi-heads $h$ while keeping the total number of Q-function estimates $N \cdot h$ fixed. We performed these experiments for all five environments, and the results are shown in Figure 6 of the Appendix. In terms of performance, we found the two best combinations to be $N = 20$, $h = 1$ and $N = 10$, $h = 2$, with the former being about 50% slower than latter in terms of computation time. In the Appendix we also consider endowing REDQ with the same multi-head ensemble architecture as AQE and find that it does not improve REDQ substantially.

In the Appendix, we also consider including a UTD ratio of $G = 5$ in both SAC and TQC, and compare these aggressive versions with AQE, also with $G = 5$. Although SAC becomes more sample efficient with $G = 5$, AQE continues to outperform both algorithms except for Humanoid, where once again TQC performs somewhat better than AQE for the final stage of training.

## 5 THEORETICAL RESULTS

In this section, we characterize how changing the size of the ensemble $N$ and the keep parameter $K$ affects the estimation bias term in the AQE algorithm. We will restrict our analysis to the tabular version of AQE (see Appendix E). Our analysis will follow similar lines of reasoning as Lan et al. (2020) and Chen et al. (2021) which extends upon the theoretical framework introduced in Thrun & Schwartz (1993).

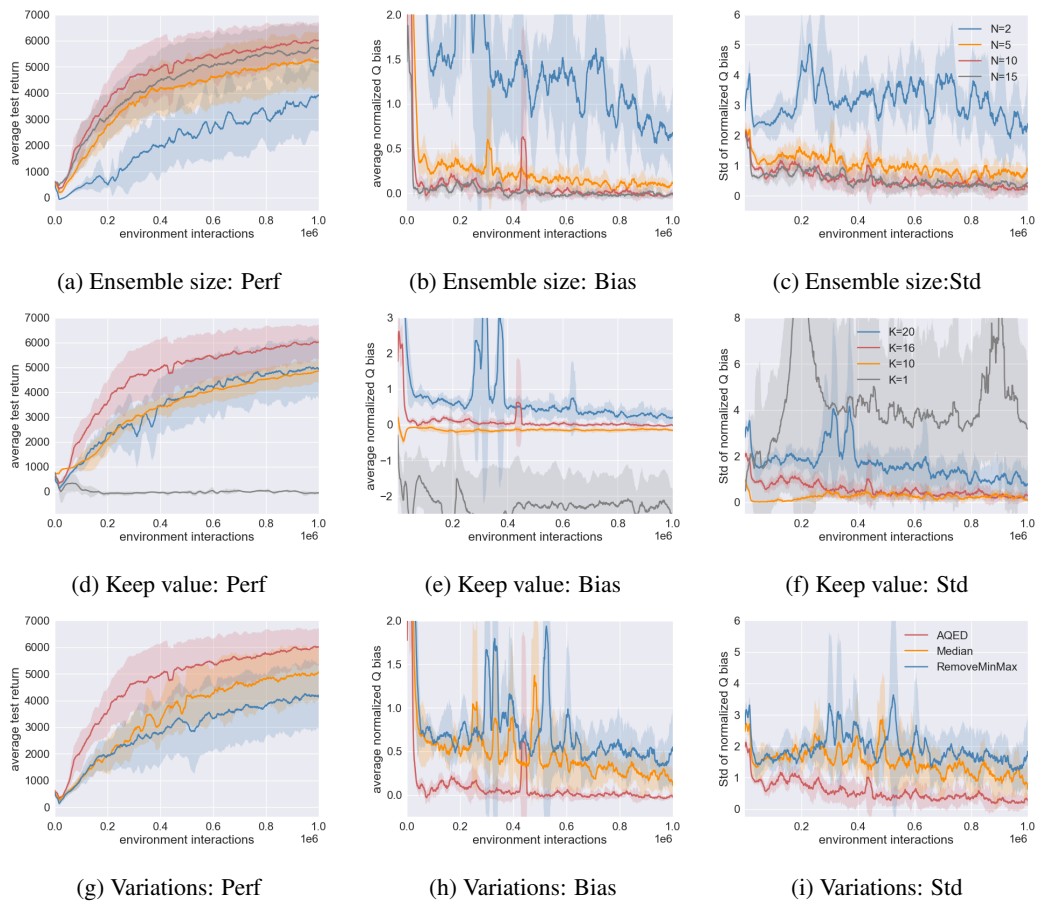

(a) Ensemble size: Perf     (b) Ensemble size: Bias     (c) Ensemble size:Std

(d) Keep value: Perf     (e) Keep value: Bias     (f) Keep value: Std

(g) Variations: Perf     (h) Variations: Bias     (i) Variations: Std

Figure 3: AQE ablation results for Ant. The top row shows the effect of the ensemble size $N$. The second row shows the effect of keep number parameter $K$. The third row compares AQE to some variants.

For each $a \in \mathcal{A}$, let $E_{K,N}(s,a)$ be the ensemble members in $\{1, \ldots, N\}$ with the $K$ lowest values of $Q^j(s,a)$, $j = 1, \ldots, N$. In the tabular case, the target for the Q networks take the form:

$$r + \gamma \max_{a'} \left( \frac{1}{K} \sum_{j \in E_{K,N}(s',a')} Q^j(s',a') \right). \tag{1}$$

Define the *post-update estimation bias* as

$$Z_{K,N} := r + \gamma \max_{a'} \left( \frac{1}{K} \sum_{j \in E_{K,N}(s',a')} Q^j(s',a') \right) - \left( r + \gamma \max_{a'} Q^\pi(s',a') \right)$$

$$= \gamma \left[ \max_{a'} \left( \frac{1}{K} \sum_{j \in E_{K,N}(s',a')} Q^j(s',a') \right) - \max_{a'} Q^\pi(s',a') \right] \tag{2}$$

Under this definition, if $\mathbb{E}[Z_{K,N}] > 0$, then the expected post-update estimation bias is positive and there is a tendency for the positive bias to accumulate during updates. Similarly, if $\mathbb{E}[Z_{K,N}] < 0$, then the expected post-update estimation bias is negative and there is a tendency for the negative bias to accumulate during updates. Ideally, we would like $\mathbb{E}[Z_{K,N}] \approx 0$

Also let

$$Q^j(s,a) = Q^\pi(s,a) + e^j(s,a) \tag{3}$$

where $e^j(s, a)$ is an independent and identically distributed error term across all $j$'s and all $a$'s for each fixed $s$. We further assume that $\mathbb{E}[e^j(s, a)] = 0$. Note that with this assumption

$$\mathbb{E}\left[\frac{1}{N}\sum_{j=1}^{N} Q^j(s', a')\right] - Q^\pi(s, a) = 0,$$

that is the pre-update estimation bias is zero. The following theorem shows how the expected estimation bias changes with $N$ and $K$:

**Theorem 1.** *The following results hold for $\mathbb{E}[Z_{K,N}]$:*

1. *$\mathbb{E}[Z_{N,N}] \geq 0$ for all $N \geq 1$.*

2. *$\mathbb{E}[Z_{K-1,N}] \leq \mathbb{E}[Z_{K,N}]$ for all $K \leq N$.*

3. *$\mathbb{E}[Z_{K,N+1}] \leq \mathbb{E}[Z_{K,N}]$.*

4. *Suppose that $e_{sa}^j \leq c$ for some $c > 0$ for all $s, a$ and $j$. Then there exists an $N$ sufficiently large and $K < N$ such that $\mathbb{E}[Z_{K,N}] < 0$.*

*Proof Sketch.* Part 1 is a result of Jensen's Inequality, and Parts 2 and 3 can be shown by analyzing how the average of the $K$ smallest ensembles changes when adding an extra ensemble model. Given the first three parts, we only need to show that $\mathbb{E}[Z_{1,N}] < 0$ to show that there exists a $K$ for a sufficiently large $N$ where the expected bias is negative. See supplementary material for full proof. □

Theorem 1 shows that we can control the expected post-update bias $\mathbb{E}[Z_{K,N}]$ through adjusting $K$ and $N$. More concretely, we can bring the bias term from above zero (i.e. over estimation) to under zero (i.e. under estimation) by decreasing $K$ and/or increasing $N$.

We note also that similar to Chen et al. (2021), we make very few assumptions on the error term $e_{s,a}$. This is in contrary to Thrun & Schwartz (1993) and Lan et al. (2020), both of whom assume that the error term is uniformly distributed.

In REDQ, a random subset of ensemble models of size $M$ is chosen and for any fixed $M$, the bias does not depend on the number of ensemble models $N$ (Chen et al., 2021). We can thus see from Figure 7 in Appendix, with $M = 2$ fixed, increasing the size of ensemble $N$ with the multi-head architecture does not necessarily help the training of REDQ. Unlike REDQ, AQE can control the bias term through both the number of ensemble models used in the average calculation $K$ and the total number of ensembles $N$, allowing for more flexibility. One other drawback for REDQ is that it ignores the estimates of all other ensemble estimates except for the minimal one in the randomly chosen set, which diminishes the power of the multiple ensemble sets. In contrast, AQE utilizes makes use of most of the ensemble models when calculating the target.

## 6 CONCLUSION

Perhaps the most important take away from this study is that a simple model-free algorithm can do surprisingly well, providing state-of-art performance at all stages of training. There is no need for a model, distributional representation of the return, or in-target randomization to achieve high sample efficiency and asymptotic performance.

The experimental results in this paper show that the AQE algorithm provides state-of-the-art performance not only when hyper-parameters are customized to individual environments (as done in the TQC paper) but also when the hyper-parameters are held fixed across environments. The ablation study shows that AQE is robust to small changes in the hyper-parameters. The theoretical results complement the experimental results, showing that the estimation bias can be controlled by either varying the ensemble size $N$ or the keep parameter $K$.

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

## A HYPERPARAMETERS AND IMPLEMENTATION DETAILS

Table 6 gives a list of hyperparameters used in the experiments. Most of AQE's hyperparameters are made the same as in the REDQ paper to ensure fairness and consistency in comparisons, except that AQE has 2-head critic networks. As compared with AQE and REDQ, TQC uses a larger critic network with 3 layers of 512 units per layer. In table 7, we report the dropped atoms $d$ for TQC and the number of Q values we keep in the ensemble to calculate the target in AQE.

| Hyperparameters | AQE | SAC | REDQ | TQC |
|---|---|---|---|---|
| optimizer | | | Adam | |
| learning rate | | | $3 \cdot 10^{-4}$ | |
| discount($\gamma$) | | | 0.99 | |
| target smoothing coefficient($\rho$) | | | 0.005 | |
| replay buffer size | | | $1 \cdot 10^6$ | |
| number of critics $N$ | 10 | 2 | 10 | 5 |
| number of hidden layers in critic networks | 2 | 2 | 2 | 3 |
| size of hidden layers in critic networks | 256 | 256 | 256 | 512 |
| number of heads in critic networks $h$ | 2 | 1 | 1 | 25 |
| number of hidden layers in policy network | | | 2 | |
| size of hidden layers in policy network | | | 256 | |
| mini-batch size | | | 256 | |
| nonlinearity | | | ReLU | |
| UTD ratio G | 5 | 1 | 5 | 1 |

Table 6: Hyperparameter values.

| Environment | Dropped atoms per critic | Kept Q values out of $N \cdot h$ values |
|---|---|---|
| Hopper | 5 | 10 |
| HalfCheetah | 0 | 20 |
| Walker | 2 | 16 |
| Ant | 2 | 16 |
| Humanoid | 2 | 16 |

Table 7: Environment-dependent hyper-parameters for TQC and AQE.

## B    ADDITIONAL RESULTS FOR AQE, SAC-5 AND TQC-5

Figure 4 presents the performance of AQE, SAC-5 and TQC-5 for all the environments. SAC-5 and TQC-5 uses UTD ratio G = 5 for SAC and TQC, respectively. We can see that AQE continues to outperform both algorithms except for Humanoid, where TQC performs somewhat better than AQE in the final stage training. SAC becomes more sample efficient with $G = 5$; however, AQE still beats SAC-5 by a large margin.

## C    ADDITIONAL RESULTS FOR PARAMETER $K$

Due to lack of space, Figure 3 only compares different AQE keep numbers $K$ for Ant. Figure 5 shows the performance, average estimation bias and standard deviation for all five environments. Consistent with the theoretical result in Theorem 1, by decreasing $K$, we lower the average bias.

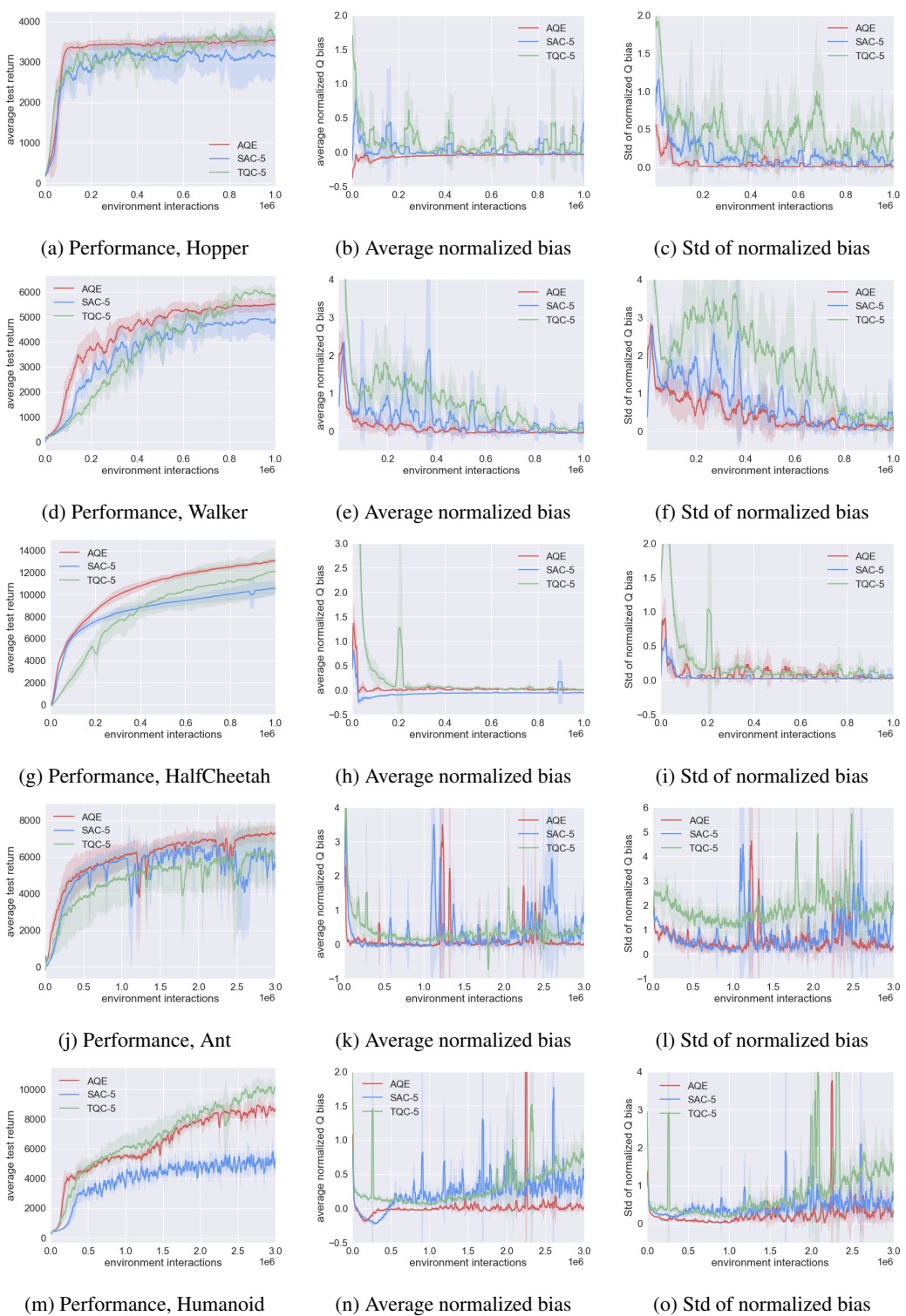

Figure 4: Performance, average and std of normalized Q bias for AQE, SAC-5 and TQC-5. All of the algorithms in this experiment use UTD = 5.

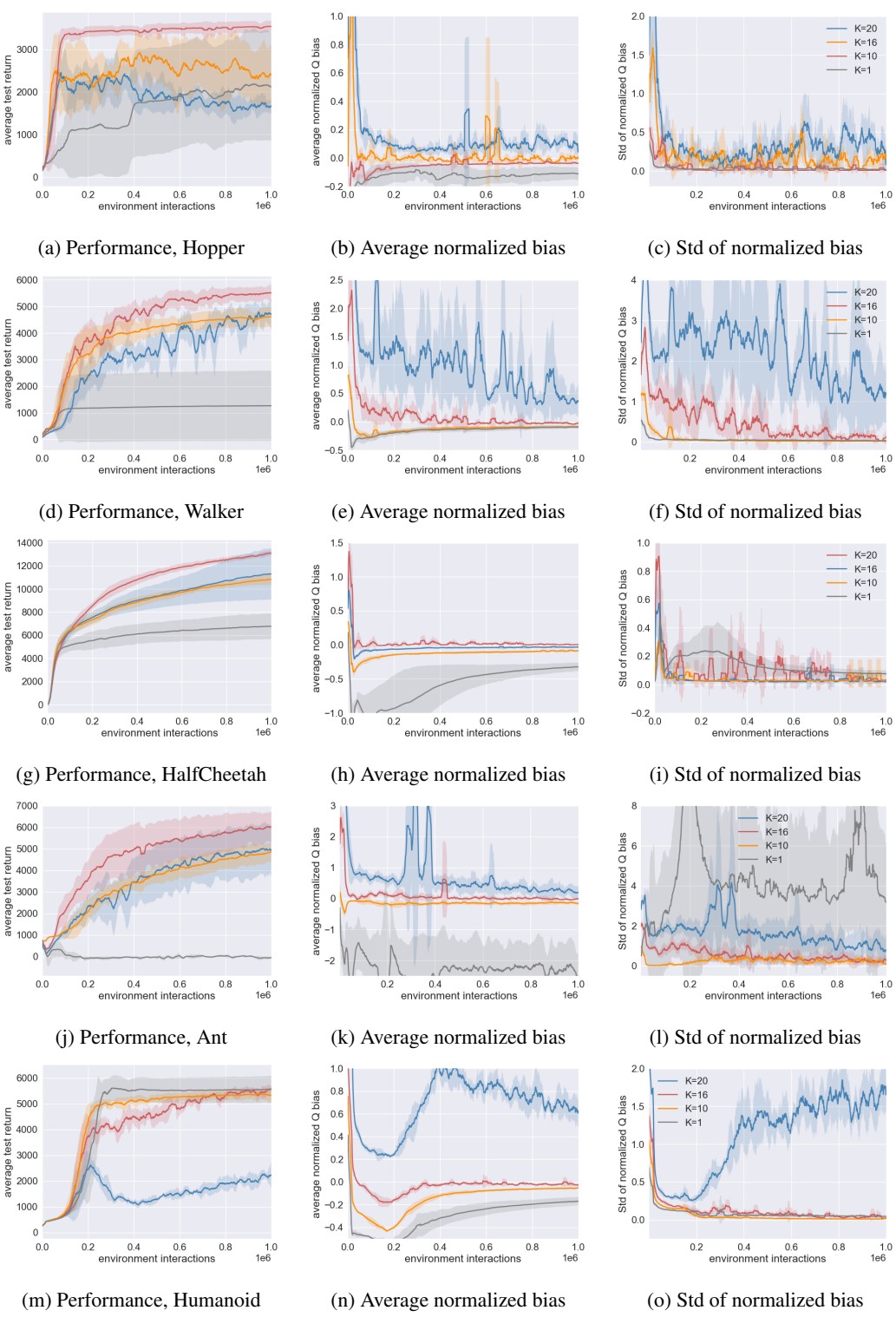

(a) Performance, Hopper     (b) Average normalized bias     (c) Std of normalized bias

(d) Performance, Walker     (e) Average normalized bias     (f) Std of normalized bias

(g) Performance, HalfCheetah     (h) Average normalized bias     (i) Std of normalized bias

(j) Performance, Ant     (k) Average normalized bias     (l) Std of normalized bias

(m) Performance, Humanoid     (n) Average normalized bias     (o) Std of normalized bias

Figure 5: Performance, average and std of normalized Q bias for AQE with different values of $K$.

# D ADDITIONAL RESULTS FOR MULTI-HEAD ARCHITECTURE

Due to lack of space, Figure 3 only compares the different size of the ensemble $N$ and the number of heads $h$ for Ant. Figure 6 shows the results for all five environments. We can see that the combination of $N = 10, h = 2$ and $N = 20, h = 1$ have comparable performance. However, $N = 10$ and $h = 2$ is faster in terms of computation time.

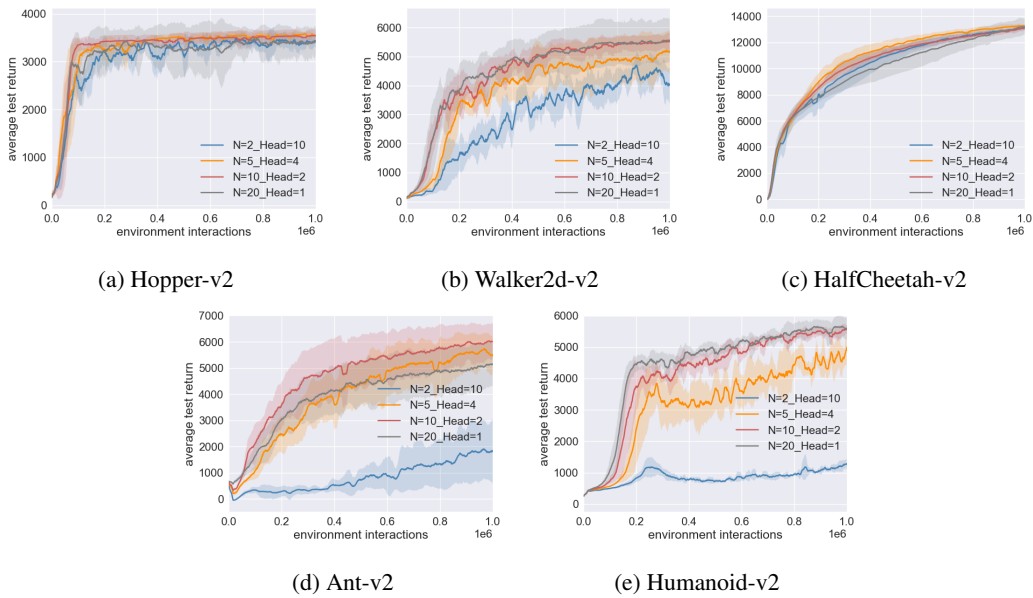

Figure 6: Performance for AQE with different combinations of number of Q networks and number of heads.

Will the performance of REDQ match that of AQE if we also provide REDQ a multi-head architecture? Figure 7 examines the performance of REDQ when it is endowed with the same multi-head architecture as AQE. We see that the performance of REDQ does not substantially improve.

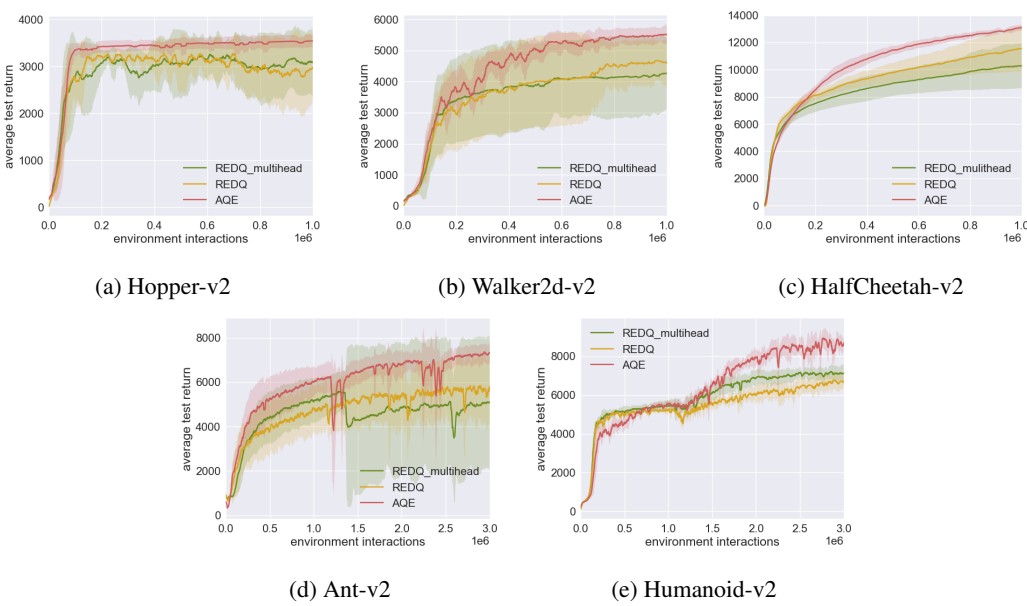

Figure 7: Performance of REDQ with N=10 and heads = 2 as compared with REDQ and AQE.

# E  TABULAR AQE WITH $N$ ENSEMBLE MEMBERS AND $d$ DROPS

---

**Algorithm 2** Tabular AQE

---

1: **Initialize:** $Q^j(s, a)$ for all $s \in \mathcal{S}$, $a \in \mathcal{A}$, $j = 1, \ldots, N$.
2: **repeat**
3:   For some state $s \in \mathcal{S}$, choose $a \in \mathcal{A}$ based on $\left\{Q^j(s, a)\right\}_{j=1}^N$, observe $r, s'$.
4:   For each $a' \in \mathcal{A}$, let $E_{K,N}(s', a')$ be the ensemble members in $\{1, \ldots, N\}$ with the $K$ lowest values of $Q^j(s', a')$, $j = 1, \ldots, N$.
5:   Get target

$$y = r + \gamma \max_{a' \in \mathcal{A}} \frac{1}{K} \sum_{j \in E_{K,N}(s', a')} Q^j(s', a')$$

6:   **for** $j = 1, \ldots, N$ **do**
7:     Update each $Q^j(s, a)$

$$Q^j(s, a) \leftarrow Q^j(s, a) + \alpha(y - Q^j(s, a))$$

8:   $s \leftarrow s'$
9: **until** end

---

# F  PROOFS

We first present the following lemma:

**Lemma F.1** (Chen et al., 2021). *Let $X_1, X_2, \ldots$ be an infinite sequence of i.i.d. random variables with cdf $F(x)$ and let $\tau = \inf x : F(x) > 0$. Also let $Y_N = \min\{X_1, X_2, \ldots, X_N\}$. Then $Y_1, Y_2, \ldots$ converges to $\tau$ almost surely.*

*Proof.* See Appendix A.2 of Chen et al. (2021) □

**Theorem 1.** *The following results hold for $\mathbb{E}[Z_{K,N}]$:*

1. $\mathbb{E}[Z_{N,N}] \geq 0$ *for all $N \geq 1$.*

2. $\mathbb{E}[Z_{K-1,N}] \leq \mathbb{E}[Z_{K,N}]$ *for all $K \leq N$.*

3. $\mathbb{E}[Z_{K,N+1}] \leq \mathbb{E}[Z_{K,N}]$.

4. *Suppose that $e_{sa}^j \leq c$ for some $c > 0$ for all $s, a$ and $j$. Then there exists an $N$ sufficiently large and $K < N$ such that $\mathbb{E}[Z_{K,N}] < 0$.*

*Proof.*   1. By definition,

$$\mathbb{E}[Z_{N,N}] = \gamma \mathbb{E}\left[ \max_{a'} \left( \frac{1}{N} \sum_{j=1}^N Q^j(s', a') \right) - \max_{a'} Q^\pi(s', a') \right]$$

$$\geq \gamma \left[ \max_{a'} E\left[ \left( \frac{1}{N} \sum_{j=1}^N Q^j(s', a') \right) \right] - \max_{a'} Q^\pi(s', a') \right] \tag{4}$$

$$= \gamma \left[ \max_{a'} Q^\pi(s', a') - \max_{a'} Q^\pi(s', a') \right] = 0$$

2. Let

$$\bar{Q}_{K,N}(s, a) = \frac{1}{K} \sum_{j \in E_{K,N}} Q^j(s, a). \tag{5}$$

Since for any state $s$, $\max_a \bar{Q}_{K+1,N}(s,a) \geq \max_a \bar{Q}_{K,N}(s,a)$,

$$
\begin{aligned}
\mathbb{E}[Z_{K+1,N}] &= \gamma \, \mathbb{E}\left[\max_a' \bar{Q}_{K+1,N}(s',a') - \max_{a'} Q^\pi(s',a')\right] \\
&\geq \gamma \, \mathbb{E}\left[\max_a' \bar{Q}_{K,N}(s',a') - \max_{a'} Q^\pi(s',a')\right] \\
&= \mathbb{E}[Z_{K,N}]
\end{aligned}
\tag{6}
$$

3. Comparing $\mathbb{E}[Z_{K,N}]$ and $\mathbb{E}[Z_{K,N+1}]$ is equivalent to comparing $\bar{Q}_{K,N}(s,a)$ and $\bar{Q}_{K,N+1}(s,a)$. Since $e^j(s,a)$ is i.i.d., by extension $Q^j(s,a)$ is also i.i.d. for $j = 1, 2, \cdots$. Suppose $Q^j(s,a)$ is drawn from some probability distribution $F$, then given $\bar{Q}_{K,N}(s,a)$, $\bar{Q}_{K,N+1}(s,a)$ can be calculated by generating an additional $Q^i(s,a)$ from $F$. The new sample $Q^i(s,a)$ affects the calculation of $\bar{Q}_{K,N+1}(s,a)$ under the following two cases:

   - If $Q^i(s,a) > \max_{j \in E_{K,N}} Q^j(s,a)$, then the lowest $K$ values remain unchanged hence $\bar{Q}_{K,N}(s,a) = \bar{Q}_{K,N+1}(s,a)$.
   - Else if $Q^i(s,a) \leq \max_{j \in E_{K,N}} Q^j(s,a)$, then $\max_{j \in E_{K,N}} Q^j(s,a)$ would be removed from and $Q^i(s,a)$ would be added to the set of lowest $K$ values, therefore $\bar{Q}_{K,N+1}(s,a) \leq \bar{Q}_{K,N}(s,a)$.

   Combining the two cases $\bar{Q}_{K,N+1}(s,a) \leq \bar{Q}_{K,N}(s,a)$, therefore $\mathbb{E}[Z_{K,N+1}] \leq \mathbb{E}[Z_{K,N}]$

4. Since $\mathbb{E}[Z_{N,N}] \geq 0$, $\mathbb{E}[Z_{K,N}] \leq \mathbb{E}[Z_{K+1,N}]$ and $\mathbb{E}[Z_{K,N+1}] \leq \mathbb{E}[Z_{K,N}]$. It is suffice to show that $\mathbb{E}[Z_{1,N}] < 0$ for some $N$. The rest of the proof largely follows Theorem 1 of Chen et al. (2021).

   Let $\tau = \inf\{x : F_a(x) > 0\}$ where $F_a(x)$ is the cdf of $Q^j(s,a)$, $j = 1, 2, \ldots$. By Lemma 1, $\bar{Q}_{1,N}(s,a) = \min_{1 \leq j \leq N} Q^j(s,a)$ converges almost surely to to $\tau_a$ for each $a$. Since the action space is finite, it then follows that $\max_a \bar{Q}_{1,N}(s,a)$ converges almost surely to to $\tau = \max_a \tau_a$. Due to our assumption that $e^j(s,a) \leq c$ and that $Q^\pi(s,a)$ is finite, it then follows that $\max_a \bar{Q}_{1,N}(s,a)$ is also bounded above. By Part 3 of the theorem, $\bar{Q}_{1,N}(s,a)$ is monotonoically decreasing w.r.t. $N$. and since $\max_a \bar{Q}_{1,N}(s,a)$ is also bounded above and converges almost surely to $\tau$, we have

$$
\begin{aligned}
\mathbb{E}[Z_{1,N}] &= \gamma \left( \mathbb{E}[\max_a \min_{1 \leq j \leq N} Q^j(s,a)] - \max_a Q^\pi(s,a) \right) \\
&= \gamma \left( \mathbb{E}[\max_a Y_a^N] - \max_a Q^\pi(s,a) \right) \stackrel{N \to \infty}{\longrightarrow} \gamma \left( \max_a \tau_a - \max_a Q^\pi(s,a) \right) < 0
\end{aligned}
\tag{7}
$$

   where the last equality follows from the assumption that the error $e^j(s,a)$ is non-trivial, and hence $\tau_a < \max_a Q^\pi(s,a)$ for all $a$. Therefore for a sufficiently large $N$, there exists a $1 \leq K \leq N$ such that $\mathbb{E}_{K,N} < 0$.

$\square$

