# OpenReview forum: "Aggressive Q-Learning with Ensembles: Achieving Both High Sample Efficiency and High Asymptotic Performance"
_ICLR.cc/2022/Conference — ICLR 2022 Submitted_

### Official Review · Reviewer_fDzQ · 2021-10-25

**Correctness:** 3
**Technical Novelty And Significance:** 3
**Empirical Novelty And Significance:** 2
**Recommendation:** 3
**Confidence:** 4

**Main Review:**

My major concern is that the empirical results are not informative at all.

As shown by Figure 5 of Henderson et al. (2018), 5 random seeds are clearly not enough for empirical comparison. Consequently, I cannot draw any conclusion from the figures or convince myself of any conclusion the authors draw from the figures. The tables are even worse since they do not contain any confidence estimation. I am disappointed in that the authors indeed cited Henderson et al. (2018) regarding the concern of reproducibility of RL research but still consider only 5 random seeds.

Moreover, the paper considers only 5 domains. The contribution of this work purely empirical so I would expect the authors to evaluate the algorithms in much more domains, e.g., in DeepMind control suite. I cannot convince myself of any conclusion with only 5 domains, especially when the performance of the proposed algorithms and the baselines are of the same order.

The theory also looks weird. The authors assume that averaged Q estimation across the ensemble is accurate (the first equation in page 9). If this assumption indeeds holds, we can simply use the average without using any method for reducing the overestimation bias. Further, under this assumption, I believe Theorem 1 is just trivial.

I do appreciate that the authors achieve good performance with simpler approaches than existing work. That being said, it might be possible to present the paper from a "rethinking" perspective.

**Summary Of The Paper:**

The paper proposes a new method for mitigating the overestimation bias in Q-learning-like algorithms. An ensemble of critics of size N is trained while only the small K individuals are used for computing the bootstrapping targets. Empirical results are provided to support the claim.

**Summary Of The Review:**

My major concern is that the empirical results are not informative at all.

---

> ### Author Response · Authors · 2021-11-20
> **Reply to your review**
>
> Thank you so much for your careful review of the paper. Below we respond to your various points.
>
> 1.We agree that more seeds are preferable.  However the computational resources available to us are limited. We also note however that it is common in other papers to use 5 seeds which do still give quite informative results (e.g. the original REDQ paper)
>
> 2.Again we would like to experiment on more domains but we are bounded by the amount of computational resources available to us. So we decided to prioritize the domains used in both the REDQ and the TQC papers.
>
> 3.The bias issue for any Q-learning-based algorithm arises from the post-update estimation (see Eq. 2), which is what is used for the Q-target, it is quite standard to assume the pre-update estimation (Eq. 3) to be unbiased (see e.g. Thrun and Schwarz (1993), Chen et al. (2021)). Even with the zero pre-update estimation assumption, part 1 of Theorem 1 of our paper shows that the post-update bias is non-negative when we simply average over all N’s.

---

### Official Review · Reviewer_iNq8 · 2021-10-26

**Correctness:** 3
**Technical Novelty And Significance:** 2
**Empirical Novelty And Significance:** 2
**Recommendation:** 6
**Confidence:** 4

**Main Review:**

Strengths:
+ source code to replicate the proposed method is provided.
+ the proposed method is simple.
+ the performance (sample efficiency) of the proposed method is good.

Weaknesses:

- Incremental improvement from the REDQ paper:

1. AQE algorithm (Algorithm 1) is almost the same as REDQ algorithm.
The main difference between AQE and REDQ is that AQE uses (1) the average of their outputs (i.e., model averaging) to target and (2) multi-head Q functions.
The idea of using multi-head functions instead of ensembles is not brand new [1].
And the idea of using averages Q-estimates as targets is not new [2].
IMO, the authors' primal contribution is to show that incorporating these into REDQ improves performance.
However, it is not clear whether the performance improvement shown in the experimental results (e.g., Figure 1) is very meaningful.
For example, is it difficult to achieve the AQE performance obtained in the experiment simply by tuning the REDQ's hyper parameters, policy and Q functions architecture?

2. the theoretical analysis (the flow of the proof of Theorem 1) is almost identical to the one done in the REDQ paper.

[1] Stefan Lee, Senthil Purushwalkam, Michael Cogswell, David J. Crandall, and Dhruv Batra. Why M
heads are better than one: Training a diverse ensemble of deep networks. CoRR, 2015.
[2] Oron Anschel, Nir Baram, and Nahum Shimkin. Averaged-DQN: Variance reduction and stabilization
for deep reinforcement learning. In Proceedings of the 34th International Conference on
Machine Learning, pp. 176–185, 2017.

- Theorem 1 and its proof are not correct:
At the beginning of pp. 18, the authors states ''Since for any state $s$,  $\max_a \bar{Q} _ {K+1, N}(s, a) \geq \max_a \bar{Q} _ {K, N} (s, a)$ .''
However, since $ \bar{Q} _ {K, N}(s, a) $ is defined as $\bar{Q} _ {K, N}(s, a) = \frac{1}{K} \sum _ {j E_ {K,N} } Q^j(s, a)$, the above is not an inequality but an equality: $\max_a \bar{Q} _ {K+1, N}(s, a) = \max_a \bar{Q} _ {K, N}(s, a)$
This means that changing the value of $K$ does not theoretically change the expected value of the estimation bias $  \mathbb{E}[Z _ {k, N}] $.
The remaining part of the proof also contains flaws.
For example,  $\bar{Q} _ {1, N} (s, a) = \min _ {1 \leq j \leq N} Q^j (s, a)$ is assumed in the proof of the 4th property of Theorem 1, but it is not consistent with Eq.5 and Algorithm 2.

Minor comments:
Algorithm 1:  Adding a more detailed explanation of how the multiple (h) estimates of each Q function are aggregated into a single scalar value in lines 10 and 14 would make the paper easier to understand.

\>\> algorithmic base, it also employs an update-to-data ratio
\>\> The REDQ paper uses G = 20 for the update-to-data ratio
update-to-data -> UTD

\>\> Hado van Hasselt. Double q-learning.
Double q-learning -> Double Q-learning

\>\> In International Conference on Machine Learning, 2020.
\>\> In International Conference on Learning Representations,  2021.
In Proceedings of ...

**Summary Of The Paper:**

In this paper, the authors show that the asymptotic performance of  REDQ is improved by replacing the minimum of Q network outputs with the average of the multi-head Q network outputs.

**Summary Of The Review:**

I recommend rejecting the paper.
A method that performs better than REDQ is beneficial from an engineering perspective.
However, as noted above, the innovation made in the paper is incremental from what made in the REDQ paper.
Also, the paper contains flaws in the theoretical analysis section that should be substantially revised.

---

> ### Author Response · Authors · 2021-11-20
> **Reply to your review**
>
> Thank you so much for your careful review of the paper. Below we respond to your various points.
>
> 1.Our parameter settings for REDQ were tuned to achieve the best performance for all environments and our results are consistent with what is presented in the REDQ paper. We are quite certain that our performance improvements are achieved by the innovations we introduced in our algorithm.
>
> 2.We have double checked our proof for Theorem 1 and we are certain it is correct. All the inequalities in our proofs are correct and we do not see any inconsistencies in our definition. We urge the reviewer to re-read our proof and in particular the definition of E_{K, N}, which refers to the minimum K Q-values out of N.
>
> 3.In short, for the Q-target calculation we used the average of the minimum K Q values and for the policy update we used the average of all N Q-values, we will make this clear in our subsequent revision.

---

> > ### Comment · Reviewer_iNq8 · 2021-11-25
> > **Response**
> >
> > Thank you for your reply and especially for correcting my misunderstanding of your theoretical analysis.
> >
> > My concern about theoretical analysis, which is one of the determinants in my overall scoring, was resolved.
> >
> > On the other hand, my concern about "Incremental improvement from the REDQ paper" is still not sufficiently resolved.
> > You argue that "our performance improvements are achieved by the innovations we introduced in our algorithm," but it is unclear whether this performance improvement is really meaningful:
> > 1. what is the significance of the performance improvement observed in the five MuJoCo environments (e.g., Figure 1)? What are the differences in the actual behaviours of the learned policies? I think it would be good to show and compare behaviours of policies produced by each method.
> > 2. there are many overlaps of shaded regions (standard deviation? or confidence interval?) in the results. So I'm not sure if these results are really reliable. (I have the same concern about the number of seeds as Reviewer fDzQ.)
> >
> > Considering the above, I have adjusted my score (R->WA).

---

### Official Review · Reviewer_DEA7 · 2021-11-01

**Correctness:** 4
**Technical Novelty And Significance:** 2
**Empirical Novelty And Significance:** 3
**Recommendation:** 5
**Confidence:** 4

**Main Review:**

 **Pros**

* The proposed AQE algorithm is simple and seems to be effective based on the presented empirical results.
* In general the paper is well written. The proposed algorithm and the empirical results are presented in a clear way.
* Empirical analysis on five MuJoCo environments shows that AQE is competitive to TQC and improves the sample efficiency performance of REDQ.
* An extensive ablation study is provided, shedding light on the impact of the three free parameters ($K, N, G$) in the performance of AQE.

**Cons and Comments**

* The novelty of this work is a little bit incremental as it can be considered as a simple modification of the Randomised Ensembled Double Q-Learning (REDQ) algorithm.
* The connection between AQE and REDQ is not clear for someone that is not familiar with the REDQ algorithm. Actually, the reader should check the paper of REDQ first to understand its connection to the AQE.
* The number of K (first row) and N (second row) should be also presented at Figure 3 that illustrates the results of the ablation study. In general, figures/legend should contain all the necessary information.
* What is the performance of AQE in the extreme case where $G=1$? According to the paper $G$ should  be greater than $1$ ($G>1$), but it would be really interesting to check the impact of updating the model multiple times on the performance of AQE.
* What is the impact of the hyper-parameter $\rho$ on the performance of AQE?
* How will the replacement of SAC with another off-policy algorithm affect the performance of AQE?

*Minor Typos*

* p. 4: infeasible in our our

**Summary Of The Paper:**

**Summary**

This work introduces Aggressive Q-learning with Ensembles (AQE), a simple model-free reinforcement learning algorithm. AQE can be considered as a modified version of the Randomised Ensembled Double Q-Learning (REDQ) algorithm. Specifically, AQE uses a number ($1 \leq K \leq N$) of the ensemble members ($N$) with the lowest Q values for calculating the target, instead of sampling a number ($1 \leq M \leq N$) of ensemble members and keeping the one with the lowest Q values as in the case of REDQ. Similarly to REDQ, AQE also employs a high update-to-data ratio ($G > 1$) to improve the sample efficiency. Empirical analyses have been conducted on five MuJoCo environments showing that the performance AQE is competitive to that of Truncated Quantile Critics (TQC) and improves the sample efficiency performance of REDQ.

**Summary Of The Review:**

In general, the proposed model-free reinforcement algorithm, called AQE, is simple and its performance is competitive with that of other baselines on five MuJoCo environments. An ablation study is also provided at the same time. My main concern as regards this work is its novelty. As aforementioned, AQE constitutes a slight variant of the already existing REDQ algorithm.

---

> ### Author Response · Authors · 2021-11-20
> **Reply to your review**
>
> Thank you so much for your careful review of the paper. Below we respond to your various points.
>
> 1.We recognize that the algorithm is similar to REDQ. However, there are some differences, as pointed out in the paper. We wish to emphasize, however, that AQE gives SOTA performance for the standard benchmark for continuous control DRL algorithms, in both early and late-stage training. We feel that it is important that community sees that SOTA performance can be achieved by a relatively simple model-free algorithm. In REDQ, a random subset of ensemble models of size M is chosen and for any fixed M, the bias does not depend on the number of ensemble models N. Unlike REDQ, we prove that AQE can control the bias term through both the number of ensemble models used in the average calculation K and the total number of ensembles N, allowing for more flexibility. One other drawback of REDQ is that in the target it uses only one (randomly chosen) ensemble member whereas AQE uses most of the ensemble members when calculating the target. REDQ is therefore diminishing the power of using an ensemble.
>
> 2.Thanks for the suggestion about Figure 3, we will add more information to illustrate Figure 3 in a revised version.
>
> 3.Thanks for the suggestion about G values. G > 1 is a main component for AQE, we will add G=1 as an ablation study.
>
> 4.The target smoothing coefficient of 0.005 is the value considered in many DRL papers such as TD3, SAC, REDQ, TQC, etc. We kept it unchanged in AQE for a fair comparison.
>
> 5.Both REDQ and TQC are based on SAC, hence for a fair comparison, we also build AQE on SAC. Replacement of SAC with other off-policy algorithms would be a new direction of research involving entropy regularization RL, which is not our primary concern.

---

### Official Review · Reviewer_q4e8 · 2021-11-02

**Correctness:** 3
**Technical Novelty And Significance:** 2
**Empirical Novelty And Significance:** 2
**Recommendation:** 3
**Confidence:** 4

**Main Review:**

[Strengths]:
1. AQE is simple and effective. Implementation requires little change to REDQ.
2. Empirical studies show that AQE achieves a much higher average return compared to SAC, TQC and REDQ in most environments.

[Weaknesses]:
1.	One of the major concerns is the similarity of this paper to a prior work REDQ. The similarity is of various aspects, including the algorithmic tables, sentences, results, analyses and even the format! On top of that, the main idea is actually very much similar, and as far as I can tell the only interesting difference seems to be the replacement of Q value from the min value of a randomly sampled set in REDQ to the average of the min K values in AQE. The authors also don’t give a convincing explanation why this replacement can improve the performance, either theoretically or intuitively.
2.	The most similar approach REDQ which the authors cited multiple times is not tested in Figure 4 of Appendix B. Compared with the baselines in that figure (SAC-5, TQC-5), REDP sounds much better in terms of reducing the Q estimation error. I am quite curious how REDQ on the bias.
3.	It’s kind of strange that a model with low bias and std provides poor performance sometimes. For example, in figure 3, the orange line (K=10) has lower bias and std than the red line (K=16) but provides worse performance. The same phenomenon appears in Figure 5 (g) (between the blue line and red line) and Figure 5 (j) (orange line and red line), etc. The authors don’t provide an explanation.
4.	Have the authors tried to adjust K dynamically, such as using a small K in the early stage to avoid overestimating and use large K in the late stage to reduce the estimation variance?

[typos]
Not sure whether or not the authors intended to say “AQE improves the sample-efficiency performance of TQC and the asymptotic performance of REDP” in the abstract? Because TQC provides SOTA asymptotic performance while REDP achieves high sample efficiency...


**Summary Of The Paper:**

This paper proposes Aggressive Q-Learning with Ensembles (AQE), which improves the sample-efficiency performance of TQC and the asymptotic performance of REDQ, thereby providing overall state-of-the-art performance during all stages of training. Empirical studies verify the effectiveness of AQE.

**Summary Of The Review:**

Overall I think the contribution of this paper is not significant to the high standard of ICLR. I am leaning to the rejection.

---

> ### Author Response · Authors · 2021-11-20
> **Reply to your review**
>
> Thank you so much for your careful review of the paper. Below we respond to your various points.
>
> 1.We recognize that the algorithm is similar to REDQ. However, there are some differences, as pointed out in the paper. We wish to emphasize, however, that AQE gives SOTA performance for the standard benchmark for continuous control DRL algorithms, in both early and late-stage training. We feel that it is important that community sees that SOTA performance can be achieved by a relatively simple model-free algorithm.
> In REDQ, a random subset of ensemble models of size M is chosen and for any fixed M, the bias does not depend on the number of ensemble models N. Unlike REDQ, we prove that AQE can control the bias term through both the number of ensemble models used in the average calculation K and the total number of ensembles N, allowing for more flexibility. One other drawback of REDQ is that in the target it uses only one (randomly chosen) ensemble member whereas AQE uses most of the ensemble members when calculating the target. REDQ is therefore diminishing the power of using an ensemble.
>
> 2.Since REDQ-5 (REDQ with UTD=5) is shown in figure 1 and figure 2, we did not put the results in figure 4 initially. We can definitely put REDQ-5 in figure 4 in a revised version.
>
> 3.We acknowledge that some ensemble methods sometimes incur a relatively lower bias and std compared to other methods in our bias-variance analysis. However, having lower bias and std in its Q estimates is not the sufficient condition for a DRL agent to learn better. The learning speed and convergence performance of DRL agents depends largely on the exploration. The algorithms considered in our analysis may differ in their efficiency of exploration, due to different exploration schemes, random initialization, etc. This may or may not lead to a difference in terms of asymptotic performance.
>
> 4.Thank you so much for the suggestion. We agree that adjusting K dynamically may help training. However, our intention is to keep the AQE algorithm simple and straightforward. Hence, we fix K in the algorithm to avoid introduction of extra hyper-parameters.

---

### Official Review · Reviewer_H9au · 2021-11-04

**Correctness:** 3
**Technical Novelty And Significance:** 3
**Empirical Novelty And Significance:** 2
**Recommendation:** 5
**Confidence:** 4

**Main Review:**


Questions:
All Line9: how is the K determined? Do you mean take K minimal Q values from the complete set of N such Q values? Why do you take K minimal Q values for computing the target? Usually the Q-learning style algorithms takes the maximum. I found this is a mysterious part of the algorithm.

Follow up the previous question, have you studied taking average from all N Q targets? The maximum of the N or K Q values, and what have you found? In the ensemble papers, people usually takes the maximum of Q values from ensembles, e.g., see eq 8 of

https://arxiv.org/abs/1811.02696

It is also interesting to see that this paper together with TQC and REDQ all uses critic ensembles. However, the above paper (ACE) uses ensembles from actors. This is an interesting alternative. It is interesting to discuss the connections and strength of both approaches.


The computation of y in line 10 of Alg. 1 uses log of the policy, this wasn’t explained. Is it a regularization effect? Why is it important for your algorithm? Have you studied this effect? (Ablation studies?)

Interestingly, in the actor update, the whole ensemble is used to average for a Q value estimate. Could you explain the choice of the complete instead of a subset of ensemble here?

The results were run for 1 million frames: this may not be able to support asymptotic behaviour. In particular, in (b)Walker2d, clearly the TQC may possibly be further faster than AQE. Could you provide more frames than shown?

In the theory part, you said we would like E(Z_KN) \appr 0. What does the approximation mean here? Do you want to minimize it? Is it positive or negative? Your theorem 1 shows this is positive, then how bigger is it than zero? Your result 2 in Th1. shows the smaller K is preferred? Is this the message? While result 3 shows when K is fixed, N should be as big as possible? How to explain this intuitively?



Minor:
The algorithm name from Q-learning is a little unconventional. There is a policy network, which is usually not called a Q-learning style algorithm.

Sec 3:
The averaging over the ensembles for Q estimates is not Polyak averaging. Polyak averaging refers to temporal averaging over the past. Here your average is nothing about time.



**Summary Of The Paper:**

629
This paper studies data efficiency and asymptotic convergence rate of deep reinforcement learning and proposed a new algorithm called AQE and claims the algorithm are both fast in both senses. The paper is mostly empirical with results on Mujoco, and compared with TQC and SAC and showed competitive performance. There is also a theorem about the update estimate bias, which tells the orderings of the bias for when K or N is increasing.

I've read the authors' feedback and other reviews. I think the paper goes hard as the asymptotic rate in experiments is not supported well. The theorem is not helping much for the two claims. There are also other problems (presentation, algorithmic discussions and comparisons) that can be much improved. Keep trying!

**Summary Of The Review:**

It appears an interesting paper, but the algorithm wasn't explained with much detail which especially didn't address critical components.

The results show sample efficiency. The asymptotic rate is not very much supported.

The theory's message isn't very clear.

---

> ### Author Response · Authors · 2021-11-20
> **Reply to your review**
>
> Thank you so much for your careful review of the paper.  Below we respond to your various points.
>
> K is a hyperparameter. AQE uses the same actor-critic framework as TD3 and SAC, where the maximization across actions takes place during the policy update (line 14 in our algorithm). Normally in SAC a minimum value is taken from an ensemble of two Q-networks. AQE, on the other hand, uses an ensemble of N Q networks and takes the average of the K minimum values. Note that if N=2 and K=1 (and G=1), then AQE is just SAC.
>
> Taking the average from all N Q values is the same as K = N. The results are shown in Figure 3 in the main body and Figure 5 in the Appendix.
>
> The log term in line 10 and line 14 is the sampled entropy for the policy pi, measuring the randomness of the policy. We use this because our algorithm is built on top of Soft-Actor-Critic(SAC) which has that log/entropy term.
>
> We use the full ensemble during policy updates to avoid double truncation. This choice is based on other papers using an ensemble of Q-functions, such as TQC, REDQ and MaxMin.
>
> Thanks for your suggestion for running more frames. In this research we run for 1 million time steps for some of the easier environments. We might consider a higher length in future research. We also note that 1 million time steps is the length considered in many DRL papers such as TD3.
>
> Sorry for the confusion about Polyak averaging. We use Polyak averaging to update Q target networks, which is shown in line 13 in Algorithm 1. We did not intend to mean the average over ensembles.

---

### Decision · Program_Chairs · 2022-01-20

**Decision:**

Reject

**Comment:**

This paper introduces a model-free RL algorithm claiming SOTA performance. All but one reviewer agreed on rejection.

#1 The empirical results are based on only 5 seeds (too low) and the plots across 5 domains show no clear evidence of improved performance due to overlapping error bars. The paper's poor empirical practice does not support the main contribution.

#2 The proposed method builds on REDQ, but the authors maintained in the response that their method performed better than REDQ (failing to articulate significant algorithmic novelty) . Even the most positive reviewer (iNq8) did not agree when the authors claimed "our performance improvements are achieved by the innovations we introduced in our algorithm". iNq8 responded "it is unclear whether this performance improvement is really meaningful". The authors never responded to iNq8's followup questions about overlapping error bars and differences in the behaviours produced by the new method.

Points #1 and #2 combine to form the clear conclusion that this work is not ready in its current from for publication.